# Will Households Invest in Safe Sanitation? Results from an Experimental Demand Trial in Nakuru, Kenya

**DOI:** 10.3390/ijerph18094462

**Published:** 2021-04-22

**Authors:** Rachel Peletz, Caroline Delaire, Joan Kones, Clara MacLeod, Edinah Samuel, Alicea Easthope-Frazer, Ranjiv Khush

**Affiliations:** 1The Aquaya Institute, P.O. Box 1603, San Anselmo, CA 94979, USA; caroline@aquaya.org (C.D.); alicea@aquaya.org (A.E.-F.); ranjiv@aquaya.org (R.K.); 2The Aquaya Institute, P.O. Box 21862-00505, Nairobi 00100, Kenya; joankones@gmail.com (J.K.); macleodclara@gmail.com (C.M.); edinah@aquaya.org (E.S.)

**Keywords:** urban sanitation, willingness to pay, latrines, Kenya

## Abstract

Unsafe sanitation is an increasing public health concern for rapidly expanding cities in low-income countries. Understanding household demand for improved sanitation infrastructure is critical for planning effective sanitation investments. In this study, we compared the stated and revealed willingness to pay (WTP) for high-quality, pour-flush latrines among households in low-income areas in the city of Nakuru, Kenya. We found that stated WTP for high-quality, pour-flush latrines was much lower than market prices: less than 5% of households were willing to pay the full costs, which we estimated between 87,100–82,900 Kenyan Shillings (KES), or 871–829 USD. In addition, we found large discrepancies between stated and revealed WTP. For example, 90% of households stated that they would be willing to pay a discounted amount of 10,000 KES (100 USD) for a high-quality, pour-flush latrine, but only 10% of households redeemed vouchers at this price point (paid via six installment payments). Households reported that financial constraints (i.e., lack of cash, other spending priorities) were the main barriers to voucher redemption, even at highly discounted prices. Our results emphasize the importance of financial interventions that address the sizable gaps between the costs of sanitation products and customer demand among low-income populations.

## 1. Introduction

Unsafe sanitation is an increasing public health concern in developing countries, particularly in low-income areas [1,2,3,4]. In addition to reducing health risks, access to improved sanitation provides further benefits of privacy, dignity, safety, and gender equity [5]. Nevertheless, in low-income urban neighborhoods, households typically rely on unimproved onsite sanitation facilities (such as pit latrines without slabs or platforms) shared by multiple households or practice open defecation [6]. For example, only 36% of the urban population in Kenya has access to private, improved facilities [6]. Understanding household demand for improved sanitation infrastructure in these settings is one critical element of urban sanitation development efforts. 

Previous efforts have compared household demand and supply costs for improved sanitation products and services. In Tanzania and Kenya, experimental trials found that <5% of rural households were willing to pay the market price for latrine slabs, though demand was much higher at discounted levels: approximately 90% of households in Kenya and 60% in Tanzania were willing to pay some amount for latrine slabs [7,8]. Similarly, previous studies have shown that WTP for safe latrine pit emptying services is substantially lower than market prices: household contributions only covered an estimated 47% of safe pit latrine emptying costs in rural Bangladesh [9], 40% of emptying costs in urban Rwanda [10], and 25–50% of emptying costs in urban Kenya [11]. This growing body of evidence indicates substantial differences between household demand and supply costs for improved sanitation products and services, which can be quantified as financial gaps [12]. Approaches for reducing these gaps include increasing the affordability of sanitation products and services, providing low-cost financing to households, and implementing subsidy schemes. A recent World Bank study, however, reports that sanitation services are generally subsidized in higher-income settings, but that subsidies do not sufficiently reach low-income households [9]. Though well-designed and targeted subsidies have the potential to improve water and sanitation service delivery to low-income households [13,14,15], accurate data on household demand are critical to estimate the amount of subsidies required.

Stated and revealed willingness to pay (WTP) methods provide options for quantifying demand for improved sanitation products and services among low-income households. Stated WTP methods generally use household surveys to assess the demand for hypothetical products and services by asking respondents to reflect on price points through a series of questions [16,17,18,19]. Survey-based WTP estimates, however, have limitations: they may be overstated due to social pressures related to prestige or due to courtesy bias; alternatively, they may be understated in attempts to keep consumer prices low. In contrast, revealed WTP methods obtain results from real money price responses (i.e., market data or experiments), and are generally considered more reliable, because they reflect respondents’ actual (rather than hypothetical) purchasing behaviors [16].

In this study, we measured the stated and revealed WTP for high-quality, pour-flush latrines, including a superstructure, among households in selected low-income areas of the city of Nakuru, Kenya. Our study addressed the following objectives: (1) to compare household demand for high-quality latrines with the costs of building these facilities and (2) to compare stated and revealed WTP by conducting both household surveys and a real money voucher experiment. We also examined why households did or did not choose to invest in high-quality, pour-flush latrines. To our knowledge, this is the first study that compares stated and revealed WTP for sanitation facilities using a real money voucher experiment.

## 2. Materials and Methods

### 2.1. Study Site

An estimated 57% of Nakuru’s population relies upon basic unlined pit latrines, and only about 36% of fecal waste generated in the city is safely collected, transported to treatment facilities, and properly disposed [20]. Approximately 57% of Nakuru’s population lives in low-income areas [21]. Our study focused on 12 of these low-income areas: Cocacola, Hilton, Imperial Tuinuane, Kamukunji, Kapkures, Kiratina, Lakeview, Lower Mithonge, Manyani, Mburu Gichua, Mzee Wanyama Rhino, and Nyamaroto (Appendix A) [21]. We selected these low-income areas based on the following five criteria: (i) located within urban or peri-urban areas of Nakuru city; (ii) did not consist of government housing or squatters illegally occupying government land; (iii) had a limited number of sewer connections (estimated < 20% coverage); and (iv) did not have ongoing sanitation provision programs.

### 2.2. Study Design

We evaluated the household demand for high-quality, pour-flush latrines through a household survey and a real money voucher trial, which were both conducted from May to December 2019 (Figure 1). After the initial household survey, we randomly distributed discount vouchers to households and measured household redemption to establish their WTP at different price points. Following the voucher trial, we conducted follow-up surveys with a subset of households to understand their voucher redemption decisions. We developed the study design and surveys based on our prior experience with conducting similar WTP studies in Kenya and Tanzania [7,8,11,22].

### 2.3. High-Quality Latrine Characteristics

To first assess the sanitation market, we conducted (i) a literature review of national and county-level policy and program documents relevant to sanitation; (ii) in-depth interviews with key water and sanitation stakeholders; (iii) transect walks in low-income areas; and (iv) focus group discussions with low-income residents (Figure 1). When selecting high-quality latrine options for this study, we considered the hygienic separation of excreta from human contact [6] and user experience (e.g., dignity, safety, privacy, etc.), acknowledging that shared sanitation facilities may be the best option in the short term in some low-income urban settings [23,24]. Based on these findings [25], we identified two high-quality latrine options as appropriate for the local context: (i) pour-flush squatting latrines connected to fully lined waste containment pits and (ii) pour-flush squatting latrines connected to the sewer. To address privacy, safety, hygiene, gender equity, cleanliness, and the needs of the elderly or disabled, both options included the following characteristics: located on the compound, a durable stone superstructure with a sheet metal tin roof, indoor tiles, a door lockable from inside and outside, an indoor solar-powered light, a handrail, a sanitary pad receptacle, and a handwashing station. We presented each latrine option as being shared by a maximum of three households [23]. We explained these sanitation options to study participants using detailed marketing graphics (Appendix A).

### 2.4. Study Population and Sampling Strategy

We targeted approximately 300 landlords or homeowners and 100 tenants for surveys. We oversampled landlords/homeowners because they are more likely to be financially responsible for sanitation [26]. Households were eligible to participate in the household survey if they met the following criteria: (i) were exclusively a residence (i.e., not a business or institution); (ii) had an adult (≥18 years) head of household available to be surveyed; and (iii) did not have high-quality sanitation, defined as an improved latrine connected to the sewer, located on the premises, and shared by a maximum of three households, for themselves or their tenants. Additionally, landlords had to live on the premises to qualify for participation.

To identify households for study participation, we first used ArcGIS mapping software (ESRI, Redlands, CA, USA) to randomly select a number of GPS coordinates proportional to each low-income area’s total population [27]. We selected separate sets of GPS coordinates to identify landlords/homeowners and tenants. Subsequently, survey enumerators navigated to each selected GPS coordinate and identified the nearest compound. Enumerators continued surveying all compounds in one designated direction until they reached a maximum of four surveys per day. If a head of household was not available during the initial visit, the enumerators returned to the household at least two more times before classifying the household as not available. In half (6/12) of the targeted urban low-income areas, we found that a large proportion of landlord/homeowners were not eligible for our survey, so enumerators exhaustively screened these areas to identify all eligible respondents. Enumerators administered the questionnaire in the local language, Kiswahili.

### 2.5. Stated Willingness to Pay

Our household survey employed the double-bounded dichotomous choice method to measure stated WTP for high-quality, pour-flush latrine options [28]. We first asked respondents to provide a yes/no answer to two closed-ended price probes (Appendix A) [18,29]. If the respondent answered yes to the first question, we then queried their WTP at a higher price. Alternatively, if the respondent answered no to the first question, we then queried their WTP at a lower amount. Other studies of WTP for water and sanitation products and services have applied similar dichotomous choice methods [9,19,22,29,30], and these methods are often used in the case of hypothetical products/services, because the sequence of questions helps respondents think about a relevant price range [17,18,19]. We followed the dichotomous choice questions with an open-ended question asking for a maximum WTP amount [31].

In each survey, we randomized the starting price points for the double-bound dichotomous choice questions (Appendix A, Table 1). We queried WTP by landlords and homeowners for the high-quality latrine options with starting price points ranging from approximately 10 to 80% of the actual costs (rounded to the nearest 10,000 KES). We estimated the actual costs of 87,100 KES (871 USD) for the pour-flush latrine connected to a lined pit and 82,900 KES (829 USD) for the pour-flush latrine connected to a sewer; both estimates included a superstructure (Table 1). Detailed figures underlying these estimates are provided in a complementary study [12]. As an alternative to large one-time payments, we also asked landlords/homeowners about their WTP in monthly installment payments, ranging from 1000 to 7000 KES (10–70 USD) (Table 1) over 12 months for the high-quality, pour-flush latrine to lined pit option. We also asked tenants about their WTP an increase in rent to support improved sanitation facilities, ranging from 200 to 800 KES (2–8 USD) per month (Table 1).

### 2.6. Revealed Willingness to Pay

Following the household survey, we conducted a real-money sales trial to estimate the revealed WTP for the high-quality, pour-flush latrine connected to a lined pit with concrete slab flooring, a ceramic pan, a pit depth of 15 feet, and the same additional features described during the household survey (i.e., made of a durable stone superstructure with a sheet metal roof, indoor tiles, a door lockable from the inside and outside, an indoor light, a handrail, a sanitary pad receptacle, and a handwashing station). To conduct our sales trial, we randomly distributed vouchers that provided discounted prices for the pour-flush latrine to eligible households, and then tracked voucher redemption to estimate the WTP at different price points. Households were eligible for vouchers if they responded to the household survey, were landlords/homeowners, and had sufficient space for construction of the pour-flush latrine on their plot (Figure 1).

We established discounted voucher price points ranging from 10,000 to 50,000 KES (100 to 500 USD) at 10,000 KES (100 USD) increments, corresponding to approximately 10–60% of the full cost of 87,100 KES (871 USD) (i.e., 40–90% discounts). The vouchers included a household identification number, a phone number for redemption, the estimated full market price, the discount amount that the household received, and the expiration date (Appendix A). Households had approximately five to six months to redeem vouchers before their expiry date of 1 December 2019. Though the voucher was distributed to one specific household, we did not restrict multiple households from combining resources; however, the majority of our respondents did not share their latrines or compounds with other households (Table 2).

To redeem their discount vouchers, households called the phone number on the voucher to coordinate the latrine construction with a mason contracted to the research project. Households paid for latrine construction in three phases: pit excavation, concrete slab flooring and ceramic pan, and superstructure. In each construction phase, households were required to make a down payment and a final payment (Appendix A). We established this system of six payments in response to household preferences for installment payments compared to large one-time outlays (Appendix A). We applied the discount levels equally to each phase and divided the payments between down and final payments; for example, if a household received an 89% discount voucher, they would receive an 89% discount on the pit excavation (normally costing 36,500 KES (365 USD)), and they would pay the remaining 11% (4191 KES, 42 USD) in two equal payments of 2095 KES (21 USD) before and after pit excavation (Appendix A). Households paid for their discounted latrines via the mobile money application M-Pesa or in cash. We reimbursed masons for the differences between the actual cost of the latrine construction and the discount voucher amount. To monitor the services provided in return for voucher redemptions, we visited all households that redeemed their vouchers at least once during latrine construction. In October 2019, approximately four months after voucher distribution, we attempted to conduct follow-up phone calls with all voucher recipients to collect preliminary feedback and ensure that respondents understood the redemption process (Figure 1).

After the vouchers had expired in December 2019, we conducted a follow-up survey with approximately one-third of the households that received discount vouchers (Figure 1). To understand redemption behaviors, we surveyed all households that redeemed vouchers and randomly selected a subset of households that did not redeem vouchers using the statistical software program R v3.6.1 [32]. To further document factors that influenced voucher redemptions, we conducted qualitative interviews with 14 households that had received a voucher offering discounted prices that were lower than their stated WTP, and with seven key informants: three community health volunteers, three staff members of the municipal provider of water and sanitation services (Nakuru Water and Sanitation Services Company Ltd. (NAWASSCO)), one mason, and one staff member of the international non-profit implementing organization, Water & Sanitation for the Urban Poor (WSUP). We selected respondents for qualitative interviews purposively to cover a range of low-income areas and respondent types (landlords, homeowners, females, and males).

### 2.7. Sample Size

Our goal was to identify at least 300 households for voucher distributions: 60 vouchers for each of the five discounted price points. This sample size allowed us to detect a minimum difference of 25 percentage points in the proportions of the population willing to purchase toilets at each discounted price point (with a statistical power of 0.8 and a significance level of 5%). We conducted follow-up surveys with approximately one-third of the study population (Figure 1), and we conducted qualitative interviews until saturation (i.e., no new information was obtained with additional data collection).

### 2.8. Data Analysis

We entered quantitative household survey responses into the CommCare survey and data management application (DiMagi Inc., Cambridge, MA, USA) on mobile phones (Samsung Galaxy J4, Korea). We conducted quality control checks on 33% (157/469) of the surveys (68 back checks and 89 spot checks). If the households were eligible but refused their vouchers, we classified them as not willing to pay the voucher price (7%, 23/334 households, Figure 1). When comparing stated and revealed WTP, we only examined the stated WTP data from households that were eligible to receive vouchers in order to compare data within the same population. To compare WTP for monthly installment payments versus one-time payments, we adjusted the 12 monthly payments to their net present value using a discount rate of 10%. We analyzed the quantitative data using the statistical software packages Stata 15 (StataCorp, College Station, TX, USA) and R v3.6.1 [32]. We analyzed the qualitative data for common themes through inductive and deductive coding using the NVivo software package (QSR International (Americas) Inc., Burlington, MA, USA). We applied the following exchange rate for the analysis: 100 KES to USD 1.00 (oanda.com (accessed on 6 June 2019)).

## 3. Results

### 3.1. Study Population

Table 2 presents the demographic details of the 469 survey respondents, 334 voucher eligible respondents, and seven voucher redeemers. All survey respondents were heads of households. The majority of survey respondents were female (67%), had at least some primary education (81%), were married (69%), and had piped water in their compound (51%). The median household size was four members. Approximately half (49%) of households were homeowners with no tenants, consistent with the median of one household per compound. The majority of households (93%) had a latrine in their compound, though it was usually unimproved (53%) or improved but used by more than three households (32%). Among households with a latrine in their compound, 43% were satisfied with their sanitation facilities. Most households (60%) had a monthly income of less than 10,000 KES (100 USD), which translates to less than 1 USD per day per person. Demographics were similar for the subsets of survey respondents that were eligible for and redeemed vouchers, though satisfaction levels with existing sanitation conditions were lower among those eligible for vouchers (31%) and voucher redeemers (0%) compared to the survey population (43%).

### 3.2. Stated WTP

Landlords/homeowners had similar stated WTP for high-quality, pour-flush latrines connected to either lined pits or the sewer: approximately 90% of households were willing to pay 10,000 KES (100 USD), but less than 5% were willing to pay 80,000 KES (800 USD) (Figure 2a). The median stated WTP was 25,000 KES (250 USD) for high-quality, pour-flush latrines connected to lined pits and 30,000 KES (300 USD) for high-quality, pour-flush latrines connected to the sewer; these numbers were identical when comparing landlords and homeowners. We found that the stated WTP for high-quality, pour-flush latrines connected to lined pits was similar when comparing payments in 12 monthly installments with one-time payments (Figure 2a). Despite the similarities in the stated WTP for the two latrine options and the two payment options, 70% of households reported a preference for high-quality, pour-flush latrines connected to lined pit latrines (compared to pour-flush latrines connected to sewerage), and 90% reported a preference for paying for sanitation options in installments (compared to a one-time payment) (Appendix A). Approximately 60% of the tenants stated that they were willing to pay some level of increased rent for having a high-quality, pour-flush latrine in their compound (Figure 2b). The median monthly rent increase that tenants were willing to pay was 200 KES (2 USD) for both latrine options; about 60% were willing to pay 200 KES (2 USD) per month, and less than 10% were willing to pay 800 KES (8 USD) (Figure 2b).

### 3.3. Revealed WTP

Only seven (or 2% (7/334)) households that were offered discount vouchers for a high-quality latrine option actually followed through with the redemption of their vouchers. All seven redeemers had received one of the three highest discount levels: three households had vouchers for 10,000 KES (100 USD, 89% discount), three households had vouchers for 20,000 KES (200 USD, 78% discount), and one household had a voucher for 30,000 KES (300 USD, 67% discount). Based on the revealed WTP data, 10% of households were willing to pay 10,000 KES (100 USD), 3% were willing to pay 20,000 KES (200 USD), 0.5% were willing to pay 30,000 KES (300 USD), and 0% were willing to pay 40,000 KES (400 USD) or 50,000 KES (500 USD) (Figure 3).

The seven households that redeemed vouchers had completed all payments and latrine construction as of July 2020. Most (6/7) households paid for the latrine from their own savings, and most (6/7) were either satisfied or very satisfied with the process for voucher redemption and latrine construction (Appendix A).

### 3.4. Exploring Low Voucher Redemption

When we followed up with one-third of the study population six months after voucher distribution (Figure 1), the majority of households that did not redeem their vouchers still had the vouchers in their possession (Table 3). Factors influencing voucher redemption are discussed below.

#### 3.4.1. Financial Constraints

Both during the follow-up household surveys and qualitative interviews, households identified financial constraints as the main barrier for voucher redemption. When households were asked why they did not redeem their voucher during the follow-up survey, 67% (68/99) of households reported that they did not have the money or had other spending priorities, and 12% (12/99) reported that the price was too high (Table 3). In subsequent qualitative interviews, households explained that they did not have the upfront cash to pay the required amounts for voucher redemption: “*The challenge is accumulating the lump sum amount of money required to build the toilet*”. Most households (82/99, 83%) also reported additional large expenditures within the last three months, such as school fees for children and family medical expenses (Table 3), which limited their ability to pay for latrine construction. For example, one household explained, “*All the money I get, I use to pay hospital bills*”. Another explained, “*I have children who have school fees and* [that I] *need to feed. I do not have a permanent job to cater to all these needs*”. When asked how households would spend extra earnings, the most common responses were contributing to savings, small businesses, or agriculture, and household construction or upgrades (Table 3).

Households expressed that paying in monthly installments would allow them to better balance sanitation expenses with other competing priorities: “*If I could pay in installments, that would be better so I can plan my expenses*”. Other respondents noted that monthly installments would “*give me time to look for money to build the toilet*”, and would “*be a lot easier because I can save a certain amount each month*”. We noted that the voucher payment system of six separate payments was effectively six installments, though this may not have provided a sufficient breakdown of payments for most households. As previously described, we also found that installment payments did not increase the total stated WTP (Figure 2a).

#### 3.4.2. Redemption Process and Trust

A few households were unclear about the household redemption process or were waiting for additional follow-up by the research team; 8% (8/99) of households were unsure how to redeem the voucher at the time of the follow-up surveys (Table 3). During the qualitative interviews, some households reported that they did not know who to call for voucher redemption, particularly if they had low levels of literacy. Instead of calling the number on the voucher, some respondents expected additional visits or follow-up calls: “*I really wanted to construct the toilet, but I waited for* [the enumerator] *to call … If you had followed up with me immediately, I would have constructed the toilet using the voucher I was given*”. Some key informants also suggested that households required additional reminders to redeem the voucher and to sensitize them to the importance of sanitation, indicating that households may have forgotten about the voucher or about the deadline to redeem.

Several households reported that they did not trust the voucher (i.e., that it was fraudulent) or the designated mason (i.e., the mason would accept the money without constructing the latrine): 5% (5/99) expressed these concerns in the follow-up surveys. One respondent stated, “*It* [is] *not easy for someone to be trusted when sending money to them*”, and other households recounted previous experiences with fraudulent offers. Some households were also skeptical as to why they received a voucher and their neighbors had not. Key informants noted that households were not familiar with the enumerators or organization (The Aquaya Institute) distributing the vouchers, and therefore were less likely to trust the voucher. Some stakeholders and households recommended involving the municipal utility (NAWASSCO) or public health officers in order to build trust (Table 3). In addition, households mentioned showcasing “*early adopters*” (i.e., latrines that were constructed as part of the program) or demonstration latrines to the rest of the community because “*most people would want to see how it* [the latrine] *is constructed before redeeming the voucher*” (Table 3).

#### 3.4.3. Decision-Making Responsibilities

We also found that a lack of communication among household members contributed to low voucher redemption levels. For example, in some instances, the voucher recipient did not fully inform other household decision-makers about the voucher. During the follow-up surveys, 7% (7/99) of households reported that they did not redeem the voucher because they were not the decision-maker, and 47% reported that a spouse or another family member had decided not to redeem (Table 3). Though households were only eligible for vouchers if they were landlords or homeowners, sometimes these households did not feel responsible for making collective financial decisions for the compound; one respondent explained that her brother was the landlord who made all of the financial decisions for the compound, but he did “*not accept to sit down and have a conversation on how to plan for the money collected from the tenants … which hindered* [her] *from redeeming the voucher*”. Furthermore, a few households reported some confusion or lack of clarify regarding plot ownership.

#### 3.4.4. Latrine Design

Most households reported that they liked the high-quality latrine designs, though a few households did claim that the design was a barrier for voucher redemption. During qualitative interviews, some households expressed preferences for a different sanitation option (e.g., septic tanks that required less frequent emptying), and several households preferred having their toilet inside the house rather than outside, where it would be shared with other households in the compound. Furthermore, some landlords preferred to have a sanitation facility with multiple doors to serve compounds with several households. Nevertheless, during the follow-up household survey, only 3% of households expressed concerns with the latrine design: 2% (2/99) preferred to make modifications to their existing latrines, and 1% (1/99) did not like the latrine dimensions (Table 3). We also found that 21% (21/99) of households reported making modifications to their existing latrine in the past six months, most commonly improving the superstructure (Table 3).

## 4. Discussion

Most low-income households in urban areas of developing countries lack access to improved sanitation that reduces exposure to fecal waste. Strategies to address these sanitation needs must consider the balance between market finance (the amount that householders pay directly or through rents) and subsidy finance (some combination of tax-based government funds, tariff-based cross-subsidies, and donor inputs) [33]. Information on the amounts that households are willing to pay directly is critical for determining the requirements for subsidy finance. In this study, we measured the amounts that low-income households in the city of Nakuru, Kenya, were willing to pay for (i) high-quality, pour-flush latrines connected to fully lined pit latrines and (ii) high-quality, pour-flush latrines connected to the sewer. We utilized two methods to determine the WTP: household surveys to quantify stated WTP and a randomized distribution of discount vouchers to quantify revealed WTP.

Our household surveys found that household stated WTP for high-quality, pour-flush latrines with an accompanying superstructure was much lower than market prices. Less than 5% of households reported that they were willing to pay the full estimated costs of between 82,900 and 87,100 KES (829–871 USD). In comparison, approximately 90% of households reported that they were willing to pay 10,000 KES (100 USD), corresponding to an 89% discount of the market price. These results are similar to findings from other studies, which have shown that demand for sanitation products and services in low-income settings is sensitive to price and often well below supply costs [7,8,9,10,11,12]. We also found that 60% of tenants were willing to pay a median rent increase of 200 KES (2 USD) per month for high-quality sanitation products, which was 10% of the median rent price (Table 2). Other studies have found that tenants are willing to pay similar amounts (2.20 USD in Lusaka, Zambia) or more (655 KES in Kisumu, Kenya) in rent increases for sanitation facilities [26,34]. In this study, households reported that financial constraints were the main reasons for low demand. Households did not prioritize investing in moving up the sanitation ladder [6]; it is important to note that the majority of respondents (93%) already had access to some type of sanitation facility, though commonly unimproved or shared among more than three households (Table 2). Most households reported that they were unable to invest in sanitation due to either a lack of cash or other spending priorities, such as school fees or medical expenses (Table 3). Similar financial restrictions were reported in other studies of demand for sanitation products [7,8]. The limited abilities of study households to invest in high-quality latrines is not surprising, given that the majority of households reported monthly incomes of 10,000 KES (100 USD) or lower (Table 2). In conclusion, low demand can also be interpreted as a general constraint of low income.

The large discrepancies between the stated and revealed WTP, however, merit further discussion. Ninety percent of the surveyed households stated that they were willing to pay the most discounted price point of 10,000 KES (100 USD, 89% discount) for a high-quality latrine with a superstructure (Figure 3). Yet, only 10% of households actually redeemed vouchers at this price point (Figure 3). Probable explanations for this difference between stated and revealed WTP include the possibilities that respondents did not fully consider budget constraints when reporting stated WTP. Most respondents reported recent large expenditures, some of which, were likely unforeseen, such as medical expenses (Table 3). Other studies have found that overstating WTP due to unexpected expenditures is common [35,36].

Furthermore, a meta-analysis of demand studies found that stated WTP inaccuracies increased with the product value: specifically, stated WTP was likely to be more inflated for high-priced products compared to lower-priced items that fell within the respondents’ budgets [35]. This may explain why, in contrast to this study, other measurements of demand for sanitation improvements found greater consistencies between stated and revealed WTP [11,34], though stated WTP may be somewhat inflated [11,37]. It is also probable that stated WTP is inflated for products that are more of a novelty or less familiar to consumers [35], which may have been the case with some of the features of the high-quality latrines that were the subject of this research.

This study is not without limitations. First, many households in the low-income areas that we targeted were ineligible for participation in the survey and voucher-based measurements of WTP for high-quality latrines (Figure 1). The largest fraction (51%) of these ineligible households were disqualified because the landlord lived offsite, and it is possible that the absence of this population segment (offsite landlords) had a significant influence on our WTP measurements. Second, it is possible that more households would have utilized their vouchers if they were provided with a redemption period beyond the five to six months offered in this study. We do note, however, that only 2% of the study households actually claimed that the redemption period was too short (Table 3). Third, the voucher redemption may have been higher if households were more familiar with the enumerators or organization distributing vouchers; however, only 5% of study households reported that trusting the voucher was a barrier to redemption (Table 3). Finally, most of our respondents were female (67%), and many may not have had primary responsibilities for household financial decisions, though there is some evidence that women in Kenya are increasingly involved in household decision-making [38].

## 5. Conclusions

Our results indicate that expanding the use of hygienic, accessible, high-quality latrines in low-income areas of Nakuru, and likely other urban centers of Kenya, will require some form of household financial assistance, such as purchase subsidies and rebates, to address the gaps between supply and demand. Without financial inputs, high-quality sanitation will remain inherently expensive and unaffordable to the poor [12]. In addition, stated measurements of demand for an expensive latrine infrastructure may not be reliable enough to provide accurate estimates of needed assistance levels. We also note that financial assistance programs will require substantial planning, investment, and coordination. For example, between 2013–2018, the municipal utility (NAWASSCO) coordinated a program offering landlords a rebate of 200 USD for every new toilet constructed in Nakuru’s low-income neighborhoods. The program eventually resulted in the construction of over 1000 pour-flush toilets [39], but our interviews with former program staff indicated that this outcome required several months of marketing and sensitization, follow-ups with households, and output-based incentives to field marketers [40]. In partnership with other stakeholders, NAWASSCO continues to explore innovative approaches to increase sanitation coverage in Nakuru, such as turning human waste into a resource [41]. In conclusion, a robust understanding of demand, external investments, and substantial program coordination are critical requirements for achieving safe and affordable sanitation for all residents of low-income areas of Nakuru, Kenya. Similar efforts and inputs are likely necessary in other developing world cities.

## Figures and Tables

**Figure 1 ijerph-18-04462-f001:**
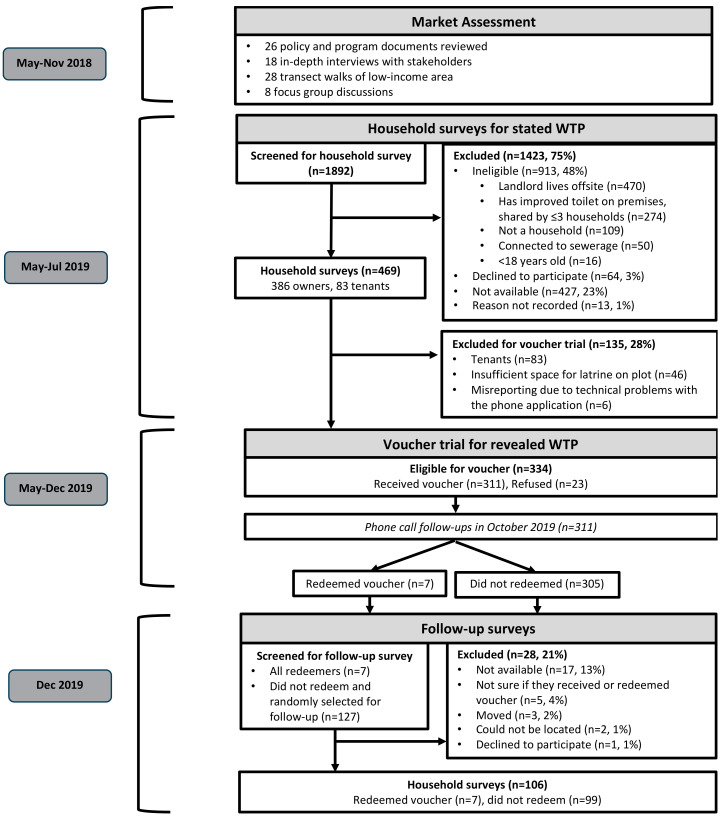
Study flow diagram.

**Figure 2 ijerph-18-04462-f002:**
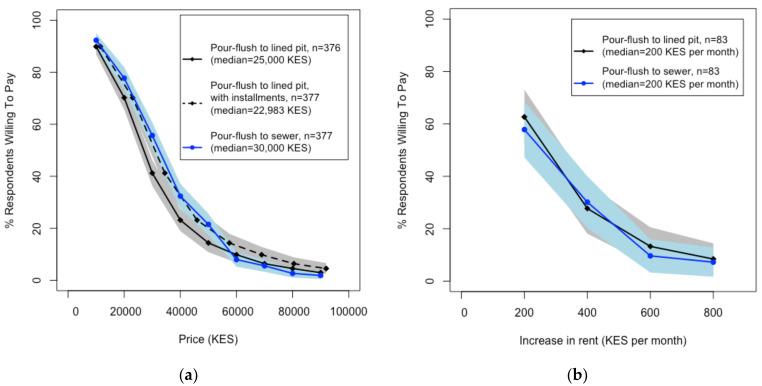
Demand curves for high-quality, pour-flush latrine options derived from survey-based stated WTP information. (**a**) Stated WTP levels among landowners and homeowners at different price points for the two high-quality latrine options. (**b**) Stated WTP levels for rent increases among tenants to support the provision of high-quality latrine options. Both groups provided WTP for pour-flush latrines connected to lined pits (black) and pour-flush latrines connected to the sewer (blue); the shaded areas represent 95% confidence intervals. In (**a**), the solid lines represent the stated WPT in the scenario of one-time payments; the dashed line represents the stated WTP in the scenario of 12 monthly installments.

**Figure 3 ijerph-18-04462-f003:**
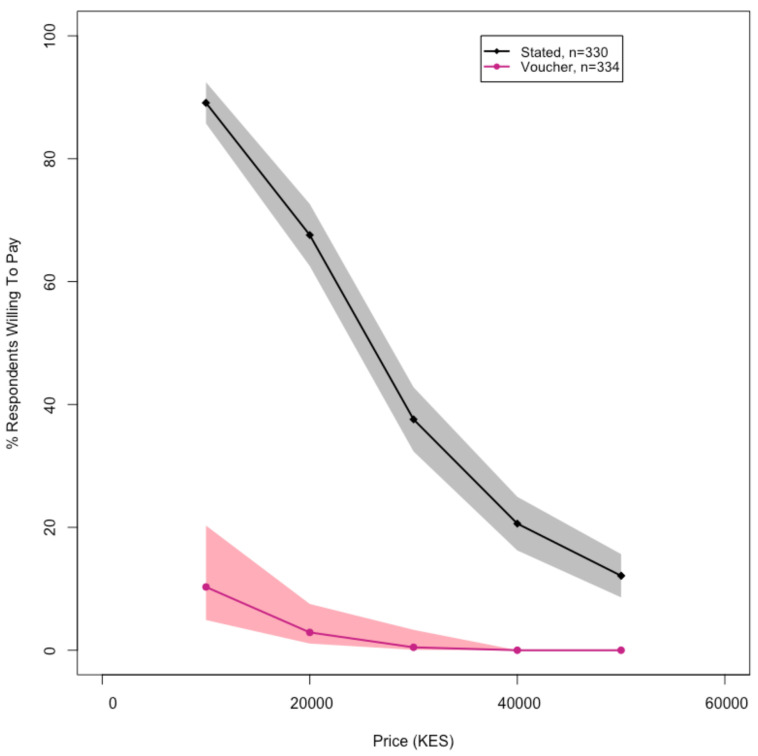
A comparison of stated (black) and revealed (pink) WTP among landowners/homeowners for high-quality, pour-flush latrines connected to a lined waste containment pit. The shaded areas represent 95% confidence intervals. For the stated WTP, the data are only from households that were eligible to receive vouchers. Of the 334 households that were eligible to receive vouchers, we have stated WTP data for 330, since four households answered “don’t know” to the stated WTP question.

**Table 1 ijerph-18-04462-t001:** Price points used in the double-bound dichotomous choice method. The starting points are listed in the last column; the lowest and highest amounts for the follow-up questions are in parentheses.

Safe Sanitation Option	Respondent ^1^	Full Price	Price Points (Ranging from 80% to 10% Discounts)
Pour-flush to lined pit (pit + slab + superstructure)	Landlords/ homeowners	87,100 KES	(5000); 10,000; 20,000; 30,000; 40,000; 50,000; 70,000; (90,000) KES
Pour-flush to sewer (sewer connection + slab + superstructure)	Landlords/ homeowners	82,900 KES	(5000); 10,000; 20,000; 30,000; 40,000; 50,000; 70,000; (90,000) KES
Pour-flush to lined pit (pit + slab + superstructure), in installments	Landlords/ homeowners	9000 KES ^1^ per month	(500); 1000; 2000; 4000; 3000; 5000; 7000; (9000) KES per month
Both pour-flush options (increased rent)	Tenants	Variable ^2^	(100), 200, 400, 600, 800, (900) KES per month

^1^ Based on the full cost of 87,100 KES, with a 20% interest rate per annum, in 12 monthly installments. ^2^ The amount of rent increase is variable, though a previous study found that rents were approximately 655 KES higher when there is sanitation [26].

**Table 2 ijerph-18-04462-t002:** Description of survey respondents, voucher eligible respondents, and voucher redeemers.

Category	All Respondents (n = 469) ^1^	Voucher Eligible Respondents (n = 334) ^2^	Voucher Redeemers (n = 7)
Gender (% female)	67%	66%	57%
Median ^3^ age	42 (32–54)	45 (34–55)	44 (30–55)
Education			
Less than primary	19%	21%	14%
Primary	38%	41%	43%
Secondary	32%	27%	29%
Above secondary	12%	11%	14%
Married or in union	69%	68%	57%
Median ^3^ household size	4 (3–6)	5 (3–6)	6 (5–6)
Presence of children under five years	40%	36%	29%
Presence of physically challenged persons	9%	12%	14%
Respondent type			
Tenants	18%	NA	NA
Homeowners with no tenants	48%	65%	57%
Landlords	34%	35%	43%
Median ^3^ number of households sharing a compound	1 (1–8)	1 (1–6)	1 (1–8)
Median ^3^ number of years lived in the compound	12 (4–20)	15 (8–24)	14 (12–22)
Primary source of water			
Piped to compound	51%	46%	43%
Piped outside compound	16%	15%	0%
Non-piped in compound	14%	16%	0%
Non-piped outside compound	19%	23%	57%
Have to pay to use primary water source	88%	87%	86%
Sanitation			
Improved latrine shared by up to three households	8%	7%	14%
Improved latrine shared by more than three households	32%	17%	29%
Unimproved latrine	53%	66%	57%
No latrine in compound	7%	10%	0%
Median ^3^ number of households sharing one latrine	1 (1–9)	1 (1–4)	1 (1–4)
Satisfied with the current sanitation ^4^	43%	31%	0%
Monthly household income (KES)			
<3000	9%	11%	0%
3000–5000	18%	18%	43%
5000–7000	13%	13%	14%
7000–10,000	20%	19%	14%
10,000–23,000	28%	27%	14%
23,000–50,000	11%	10%	14%
>50,000	2%	2%	0%
About landlords			
n	159	117	3
Median ^3^ number of dwellings owned	9 (5–14)	9 (5–14)	7 (7–17)
Median ^3^ monthly rent collected (KES)	10,000 (4900–23,000)	10,000 (4000–22,000)	15,000 (11,700–28,500)
About tenants		NA	NA
n	83		
Median ^3^ monthly rent paid (KES)	2000 (1200–2500)		

^1^ Data missing for age (1), education (1), married (1), water source (5), having to pay for primary water source (9), sanitation facility (2), number of households sharing sanitation (10), monthly household income (89), the landlord’s collected monthly rent (24), and tenant’s paid rent (2). ^2^ Data missing for age (1), education (1), married (1), primary water source (3), having to pay for primary water source (5), number of households sharing sanitation (6), monthly household income (41), and the landlord’s collected monthly rent (15). ^3^ Medians are provided with the interquartile range. ^4^ Only households with a latrine in their compound were asked this question. NA = Not applicable.

**Table 3 ijerph-18-04462-t003:** Households that did not redeem vouchers.

	Number (n = 99)	Percentage
Voucher status		
Still have it ^1^	85	86%
Lost it	5	5%
Gave it away	1	1%
Don’t know	8	8%
Who decided not to redeem		
Male respondent	17	17%
Female respondent	48	48%
Male spouse	26	26%
Female spouse	3	3%
Another family member	18	18%
Nobody (waiting for follow up)	7	7%
Why they didn’t redeem the voucher		
Didn’t have the money/other spending priorities	68	67%
Price was too high	12	12%
Unsure how to redeem	8	8%
Not my decision	7	7%
Thought voucher was fraudulent	5	5%
Other ^2^	18	18%
Spent money on latrine construction/upgrades in last six months		
No	76	77%
Modifications ^3^ to existing latrine	21	21%
Started building new dry pit latrine	2	2%
Big expenditures in the past three months		
School fees	42	42%
Medical expenses	39	39%
Ceremonies (funeral, marriage, initiation)	14	14%
House construction/repairs	11	11%
Water	6	6%
Fertilizer/agriculture	5	5%
Other ^4^	19	19%
None	17	17%
Household spending priorities if they earned an extra 1500 KES per week		
Savings	49	49%
Small business/Agriculture	41	41%
Household construction/upgrades	29	29%
Food	23	23%
School fees	21	21%
Healthcare	10	10%
Clothes	6	6%
Other ^5^	21	18%
Recommendations to increase voucher redemption ^6^		
Conduct more follow-ups/reminders	32	35%
Build demonstration latrine	32	34%
Provide more time for redemption	24	26%
Provide more explanation of sanitation benefits	23	25%
Convene community meetings	15	16%
Lower the price or provide for free	11	12%
Allow for monthly installments	10	11%
Involve the municipal utility (NAWASSCO) or public health officers	5	5%
Other ^7^	8	9%

^1^ Enumerators observed vouchers to verify that households still had the vouchers. ^2^ Other reasons for not redeeming (all < 5 households): no space for latrine, no need for a new latrine, issues with plot ownership, inadequate time to redeem, did not like latrine dimensions, wanted the latrine at another home, already started building a latrine, confusion regarding redemption procedures, and preferred modifications to existing latrine (such as adding doors or concrete flooring and improving the superstructure). ^3^ The most common modification was improving the superstructure (11 households); other modifications included adding doors, a pour-flush pan, tiles, and/or concrete flooring. ^4^ Other big expenditures included (all < 5 households): latrine construction/upgrades, business investments, household furniture, food, repaying loans/debts, court fees, land purchase, motorbike purchase, electricity bills, and veterinary fees. ^5^ Other spending priorities included furniture, water, and sanitation. ^6^ Data missing for six households. ^7^ Other recommendations for increasing voucher redemption included (all < 5 households): explaining the redemption process better and improvements in the redemption process.

## Data Availability

Data available on request due to privacy and ethical restrictions.

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
