# Peer review of "Will Households Invest in Safe Sanitation? Results from an Experimental Demand Trial in Nakuru, Kenya"

_ijerph, 2021, doi:10.3390/ijerph18094462_

Round 1
Reviewer 1 Report
Introduction:
- Page 1 lines 29-30: define unimproved sanitation
- Page 1 line 31: Why is this being framed only as a demand-side issue? Water and sanitation in rich countries is certainly a supply side responsibility. This framing needs to be revised (or at least there needs to be some explanation that part of the issue is the demand). As it is currently it is only framed as a demand side issue.
- Page 1 lines 40-41: what are the remaining questions? I think the authors need to develop the introduction and set up a bit more as it assumes the reader knows too much as it is currently written.
- There is no background information in the introduction about the scope of the problem in Kenya/Nairobi, where the study was conducted. This needs to be included.
- The authors need to discuss why they distinguish between “high-quality” sanitation and other forms of sanitation. Does this matter for health (or other outcomes)? Reading ahead, only 7% of the total study sample had no toilet, while 10% of voucher eligible respondents had no toilet. So then is this study about upgrading to “high quality” toilets? Or is it about understanding demand for any toilet? This point needs to be clarified.
- Introduction: Overall, the introduction needs to be more thorough. The research needs to be placed in global and local context. The issues with household demand need to be more clearly stated, along with the methodological studies. Furthermore, there is nothing in the introduction about the significance/novelty of this particular study.
Materials & Methods:
- Section 2.2: The study design needs to have more detail in terms of whether this is based on previous studies or it is totally new. If the former, then it should be cited appropriately.
- Section 2.3: Why wasn’t this just based on the WHO JMP definitions of improved toilets? Were these toilets shared or just for one household?
- Can sections 2.5 and 2.6 be consolidated in to the study design section?
- Line 162-164: How is 10,000 to 50,000 KES 40% to 90% of the full 87,100 KES? This is a bit confusing as those price points do not match up with the percentages.
- Line 188 – 200: There seems to be a qualitative component to this study as well. This should be separated in to a different section.
- Section 2.7: Sample size only refers to that for the quantitative study? What about the qualitative study (it is stated in the previous section, but this makes it a bit hard to follow).
- Lines 282-285: How are these percentages calculated? I’m not sure what these correspond to. Are these meant to be extrapolations to the entire population?
Discussion:
- There should be some discussion of the fact that the stated and revealed WTPs were for households that already have a toilet.
- The point about people being willing to pay more for rent is not discussed at all, which is a big omission. This is an interesting finding that needs to be explored more in the discussion as it could suggest that households want better toilets but don’t want to be the ones responsible for building them (which is not dissimilar from how people live in rich countries. You have a toilet included with your rent when you occupy an apartment or a home).
- One of the limitations of the study needs to be the offered prices seeing as though in some cases, these were many multiples of a household’s monthly income.
- In fact, that whole point needs to be further discussed, that sanitation is inherently expensive and the ultra-poor generally cannot afford it on their own. There is plenty of literature stating this from other countries (see papers from India, for example).
Author Response
Reviewer1
Introduction:
- Page 1 lines 29-30: define unimproved sanitation
- Thank you for the comments. We have revised accordingly:
- Lines 29-31: “households typically rely on unimproved onsite sanitation facilities (such as pit latrines without slabs or platforms) that are shared by multiple households or practice open defecation”
- Page 1 line 31: Why is this being framed only as a demand-side issue? Water and sanitation in rich countries is certainly a supply side responsibility. This framing needs to be revised (or at least there needs to be some explanation that part of the issue is the demand). As it is currently it is only framed as a demand side issue.
- We agree that sanitation is not only a demand-side issue, and have revised as follows:
- Lines 32-34: “Understanding household demand for improved sanitation infrastructure in these settings is one critical element of urban sanitation development efforts.”
- Page 1 lines 40-41: what are the remaining questions? I think the authors need to develop the introduction and set up a bit more as it assumes the reader knows too much as it is currently written.
- We have removed the reference to remaining questions and revised as follows:
- Lines 50-53: “Though well-designed, targeted subsidies have the potential to improve water and sanitation service delivery to low-income households [10–12], accurate data on household demand are critical to estimate the amount of subsidies required.”
- We have also added details of prior sanitation demand studies to the introduction:
- Lines 35-43: “Previous efforts have compared household demand and supply costs for improved sanitation products and services. In Tanzania and Kenya, experimental trials found that < 5% of rural households were willing to pay the market price for latrine slabs, though demand was much higher at discounted levels: approximately 90% of households in Kenya and 60% in Tanzania were willing to pay some amount for latrine slabs [7,8]. Similarly, previous studies have shown that WTP for safe latrine pit emptying services is substantially lower than market prices: household contributions only covered an estimated in 47% of safe pit latrine emptying costs in rural Bangladesh [9], 40% of emptying costs in urban Rwanda [10], and 25-50% of emptying costs in urban Kenya [11].”
- There is no background information in the introduction about the scope of the problem in Kenya/Nairobi, where the study was conducted. This needs to be included.
- We added additional details on sanitation in Kenya to the introduction:
- Lines 31-32: “For example, only 36% of the urban population in Kenya has access to private, improved facilities [6].”
- Additionally, we provide additional details on the sanitation conditions of the study site, Nakuru, Kenya, at the beginning of the materials and methods section:
- Lines 75-78: “An estimated 57% of Nakuru’s population relies upon basic unlined pit latrines, and only about 36% of fecal waste generated in the city is safely collected, transported to treatment facilities, and properly disposed [20]. Approximately 57% of Nakuru’s population lives in low-income areas [21].”
- The authors need to discuss why they distinguish between “high-quality” sanitation and other forms of sanitation. Does this matter for health (or other outcomes)? Reading ahead, only 7% of the total study sample had no toilet, while 10% of voucher eligible respondents had no toilet. So then is this study about upgrading to “high quality” toilets? Or is it about understanding demand for any toilet? This point needs to be clarified.
- We have added more about “high quality” sanitation in the methods. There is also more information on the need to examine high-quality shared sanitation provided in the references (WHO/UNICEF JMP 2017 report, and a policy brief from Water & Sanitation for the Urban Poor).
- Lines 99-103: "When selecting high-quality latrine options for this study, we considered options that hygienically separated excreta from human contact [6] and user experience (e.g., dignity, safety, privacy, etc), acknowledging that shared sanitation facilities may be the best option in the short term in some low-income urban settings [23,24].”
- We have acknowledged that this study is about moving up the sanitation ladder, as part of the discussion:
- Lines 444-447: “Households did not prioritize investing in moving up the sanitation ladder [6]: it is important to note that the majority of respondents (93%) already had access some type of sanitation facility, though commonly unimproved or shared among more than three households (Table 2).”
- Introduction: Overall, the introduction needs to be more thorough. The research needs to be placed in global and local context. The issues with household demand need to be more clearly stated, along with the methodological studies. Furthermore, there is nothing in the introduction about the significance/novelty of this particular study.
- As noted above, we added details of prior sanitation demand studies to the introduction:
- Lines 35-43: “Previous efforts have compared household demand and supply costs for improved sanitation products and services. In Tanzania and Kenya, experimental trials found that < 5% of rural households were willing to pay the market price for latrine slabs, though demand was much higher at discounted levels: approximately 90% of households in Kenya and 60% in Tanzania were willing to pay some amount for latrine slabs [7,8]. Similarly, previous studies have shown that WTP for safe latrine pit emptying services is substantially lower than market prices: household contributions only covered an estimated in 47% of safe pit latrine emptying costs in rural Bangladesh [9], 40% of emptying costs in urban Rwanda [10], and 25-50% of emptying costs in urban Kenya [11].”
- To highlight the study’s novelty, we added:
- Lines 70-72: “To our knowledge, this is the first study that compares stated and revealed WTP for sanitation facilities using a real-money voucher experiment.”
Materials & Methods:
- Section 2.2: The study design needs to have more detail in terms of whether this is based on previous studies or it is totally new. If the former, then it should be cited appropriately.
- We have clarified that the study design is based on previous studies:
- Lines 92-94: “We developed the study design and surveys based on our prior experience conducting similar WTP studies in Kenya and Tanzania [7,19–21].”
- Section 2.3: Why wasn’t this just based on the WHO JMP definitions of improved toilets? Were these toilets shared or just for one household?
- We have added the following statement to explain high-quality sanitation. There is also more information on the need to examine high-quality shared sanitation provided in the references (WHO/UNICEF JMP 2017 report, and a policy brief from Water & Sanitation for the Urban Poor).
- Lines 99-103: “When selecting high-quality latrine options for this study, we considered the hygienic separation of excreta from human contact [6] and user experience (e.g., dignity, safety, privacy, etc), acknowledging that shared sanitation facilities may be the best option in the short term in some low-income urban settings [23,24].”
- We have revised the text to clarify latrine sharing:
- Lines 110-111: “We presented each latrine option as being shared by a maximum of three households [23].”
- Can sections 2.5 and 2.6 be consolidated in to the study design section?
- We prefer to leave sections 2.5 and 2.6 separate – these provide details on the WTP methods, and the study design section provides an overview.
- Line 162-164: How is 10,000 to 50,000 KES 40% to 90% of the full 87,100 KES? This is a bit confusing as those price points do not match up with the percentages.
- Thank you for bringing this up. We have revised the text to clarify:
- Lines 181-183: “We established discounted voucher price points ranging from 10,000 to 50,000 KES (100 to 500 USD), at 10,000 KES (100 USD) increments, corresponding to approximately 10-60% of the full cost of 87,100 KES (871 USD) (i.e., 40-90% discounts).”
- Line 188 – 200: There seems to be a qualitative component to this study as well. This should be separated in to a different section.
- Correct, we have clarified this in the methods:
- Lines 214- 221: “To further document factors that influenced voucher redemptions, we conducted qualitative interviews with 14 households that had received voucher offering discounted prices that were lower than their stated WTP, and seven key informants: three community health volunteers, three staff members of the municipal provider of water and sanitation services, Nakuru Water and Sanitation Services Company Ltd (NAWASSCO), one mason, and one staff member of the international non-profit implementing organization, Water & Sanitation for the Urban Poor (WSUP).”
- We prefer to combine the qualitative and quantitative data when examining low voucher redemption, as the qualitative findings help interpret our quantitative results (section 3.4).
- Section 2.7: Sample size only refers to that for the quantitative study? What about the qualitative study (it is stated in the previous section, but this makes it a bit hard to follow).
- We have added
- Lines 228-230: “We conducted follow-up surveys with approximately one-third of the study population (Figure 1), and we conducted qualitative interviews until saturation (i.e., no new information was obtained with additional data collection).”
- Lines 282-285: How are these percentages calculated? I’m not sure what these correspond to. Are these meant to be extrapolations to the entire population?
- These percentages are of the households that were surveyed at follow-up. We have tried to clarify this in the text:
- Lines 228-229: “We conducted follow-up surveys with approximately one-third of the study population (Figure 1),”
- Lines 321-323: “When we followed up with one-third of the study population six months after voucher distribution (Figure 1), the majority of households that did not redeem their vouchers still had their vouchers in their possession (Table 3).”
Discussion:
- There should be some discussion of the fact that the stated and revealed WTPs were for households that already have a toilet.
- We have clarified that most households in our study already had a toilet:
- Lines 444-447: “Households did not prioritize investing in moving up the sanitation ladder [6]: it is important to note that the majority of respondents (93%) already had access some type of sanitation facility, though commonly unimproved or shared among more than three households (Table 2).”
- The point about people being willing to pay more for rent is not discussed at all, which is a big omission. This is an interesting finding that needs to be explored more in the discussion as it could suggest that households want better toilets but don’t want to be the ones responsible for building them (which is not dissimilar from how people live in rich countries. You have a toilet included with your rent when you occupy an apartment or a home).
- We agree that rent increases are an important point, and we have added to the discussion:
- Lines 438-441: “We also found that 60% of tenants were willing to pay a median rent increase of 200 KES (2 USD) per month for high-quality sanitation products, which was 10% of the median rent price (Table 2). Other studies have found that tenants are willing to pay similar amounts (2.20 USD in Lusaka, Zambia) or more (655 KES in Kisumu, Kenya) in rent increases for sanitation facilities [25,35].”
- One of the limitations of the study needs to be the offered prices seeing as though in some cases, these were many multiples of a household’s monthly income.
- We agree that households were not able to pay the high prices, but we do not believe this is a limitation of the study. For the voucher trial, households were offered 40-90% discounts. We state
- Lines 450-453: “The limited abilities of study households to invest in high-quality latrines is not surprising given that the majority of households reported monthly incomes of 10,000 KES (100 USD) or lower (Table 2).”
- In fact, that whole point needs to be further discussed, that sanitation is inherently expensive and the ultra-poor generally cannot afford it on their own. There is plenty of literature stating this from other countries (see papers from India, for example).
- We agree that sanitation is inherently expensive and unaffordable to the poor. We have added:
- Lines 492-493: “Without financial inputs, high-quality sanitation will remain inherently expensive and unaffordable to the poor [12].”
Reviewer 2 Report
This study compared stated and revealed willingness-to-pay (WTP) for high-quality pour-flush latrines among households in low-income areas, and employed the double-bounded dichotomous choice method to measure stated WTP for high-quality pour-flush latrine options in the city of Nakuru, Kenya. Data analysis are straightforward. However, some Minor modifications are necessary for this paper to be accepted for publication.
(1) The part of introduction may be adjusted and more concise. It will be better to add the literature review about willingness-to-pay in safe sanitation.
(2) In discussion part, it’s will be more convincing to Discuss which of the social factors can influence factor willingness to pay more.
Author Response
Reviewer 2
This study compared stated and revealed willingness-to-pay (WTP) for high-quality pour-flush latrines among households in low-income areas, and employed the double-bounded dichotomous choice method to measure stated WTP for high-quality pour-flush latrine options in the city of Nakuru, Kenya. Data analysis are straightforward. However, some Minor modifications are necessary for this paper to be accepted for publication.
(1) The part of introduction may be adjusted and more concise. It will be better to add the literature review about willingness-to-pay in safe sanitation.
- Thank you for the comments. We have added references to the literature in the introduction, also based on feedback from other reviewers:
- Lines 35-43: “Previous efforts have compared household demand and supply costs for improved sanitation products and services. In Tanzania and Kenya, experimental trials found that < 5% of rural households were willing to pay the market price for latrine slabs, though demand was much higher at discounted levels: approximately 90% of households in Kenya and 60% in Tanzania were willing to pay some amount for latrine slabs [7,8]. Similarly, previous studies have shown that WTP for safe latrine pit emptying services is substantially lower than market prices: household contributions only covered an estimated in 47% of safe pit latrine emptying costs in rural Bangladesh [9], 40% of emptying costs in urban Rwanda [10], and 25-50% of emptying costs in urban Kenya [11].”
(2) In discussion part, it’s will be more convincing to Discuss which of the social factors can influence factor willingness to pay more.
- The goal of this study was to conduct a rigorous comparison of stated and revealed WTP for high-quality toilets in the city of Nakuru, Kenya. Because these two methods for measuring demand generated different demand estimates, we did not also include an analysis of social factors associated with WTP. We note that have conducted this type of analysis in previous studies where we only applied a single method to measure WTP, and we provide these references below:
- Peletz, R.; Kisiangani, J.; Ronoh, P.; Cock-esteb, A.; Chase, C.; Khush, R.; Luoto, J. Assessing the Demand for Plastic Latrine Slabs in Rural Kenya. J. Trop. Med. Hyg. 2019, 10, 555–565.
- Peletz, R.; Cock-Esteb, A.; Ysenburg, D.; Haji, S.; Khush, R.; Dupas, P. Supply and Demand for Improved Sanitation: Results from Randomized Pricing Experiments in Rural Tanzania. Sci. Technol. 2017, acs.est.6b03846.
- Acey, C.; Kisiangani, J.; Ronoh, P.; Delaire, C.; Makena, E.; Norman, G.; Levine, D.; Khush, R.; Peletz, R. Cross-subsidies for improved sanitation in low income settlements: Assessing the willingness to pay of water utility customers in Kenyan cities. World Dev. 2019, 115.
Reviewer 3 Report
Relevant study conducted by Peletz et al. In my point of view, this is a well structured manuscript which addresses very well the emergent topic about the sanitation state of a low-income region (Nakuru, Kenya).
I have minor suggestions before it can be accepted for publication:
More data regarding the sanitation conditions in Kenya should be provided in the Introduction.
Line 74: The authors should provide more details about the applied surveys. How they constructed the surveys? Were the surveys already validated, used in previous investigations?
Author Response
Reviewer 3
Relevant study conducted by Peletz et al. In my point of view, this is a well structured manuscript which addresses very well the emergent topic about the sanitation state of a low-income region (Nakuru, Kenya).
I have minor suggestions before it can be accepted for publication:
More data regarding the sanitation conditions in Kenya should be provided in the Introduction.
- Thank you for the comments. We added additional details on sanitation in Kenya to the introduction:
- Lines 31-32: “For example, only 36% of the urban population in Kenya has access to private, improved facilities [6].”
- Additionally, we provide additional details on the sanitation conditions of the study site, Nakuru, Kenya, at the beginning of the materials and methods section:
- Lines 75-78: “An estimated 57% of Nakuru’s population relies upon basic unlined pit latrines, and only about 36% of fecal waste generated in the city is safely collected, transported to treatment facilities, and properly disposed [20]. Approximately 57% of Nakuru’s population lives in low-income areas [21].”
Line 74: The authors should provide more details about the applied surveys. How they constructed the surveys? Were the surveys already validated, used in previous investigations?
- We have added details about background work:
- Lines 92-94: “We developed the study design and surveys based on our prior experience conducting similar WTP studies in Kenya and Tanzania [7,8,11,22].”
Reviewer 4 Report
This paper investigates the WTP for implementing sanitation systems, in particular, high-quality pure flush latrines. The topic is very important and the research in this field seems to be innovative.
Overall, I found the paper informative. The objectives are clear, the research procedure is sound, and the results look reasonable. The conclusion is also supported well with the provided data.
I have two conceptional comments, which I invite the authors to note.
First, I wish to attract the authors' attention to the concept of sustainability. In lines 31-32, the authors have wisely mentioned that "understanding household demand for improved sanitation infrastructure in these settings is critical for informing urban development efforts." I think right here it is worthy to discuss the indexes that are designed to understand these demands and evaluate if a system is sustainable for a community or not. Here are some published papers, sorted based on publishing time, that I invite the authors to review.
2020: https://doi.org/10.3390/su12176937
2015: https://doi.org/10.3390/su71114537
2015: https://doi.org/10.13140/RG.2.1.2194.5763
1999: https://doi.org/10.2166/wst.1999.0244
In the introduction section, I suggest you discuss and in particular focus on the economic evaluation aspects of these indexes.
The second concept is about the core of this paper, WTP. This study is focusing on the implementation of flush latrines. Again the authors have wisely discussed the role of "providing more explanation of sanitation benefits" on the "recommendations to increase voucher redemption." This is great. However, implementing a sanitation system that can give benefits to the community as well as the background of the community to utilize the human sanitary wastes as fertilizer may also be useful in increasing the WTP. In another word, the user may be willing to pay more if s/he knows that the system can provide him/her direct economic benefits.
In this regard, I wish to invite the authors to check the papers listed below regarding the direct benefit that a sanitation system can give to the community. This concept might be worthy to be discussed in the discussion section.
https://doi.org/10.1177/0734242X10390073
Author Response
Reviewer 4
This paper investigates the WTP for implementing sanitation systems, in particular, high-quality pure flush latrines. The topic is very important and the research in this field seems to be innovative.
Overall, I found the paper informative. The objectives are clear, the research procedure is sound, and the results look reasonable. The conclusion is also supported well with the provided data.
I have two conceptional comments, which I invite the authors to note.
First, I wish to attract the authors' attention to the concept of sustainability. In lines 31-32, the authors have wisely mentioned that "understanding household demand for improved sanitation infrastructure in these settings is critical for informing urban development efforts." I think right here it is worthy to discuss the indexes that are designed to understand these demands and evaluate if a system is sustainable for a community or not. Here are some published papers, sorted based on publishing time, that I invite the authors to review.
2020: https://doi.org/10.3390/su12176937
2015: https://doi.org/10.3390/su71114537
2015: https://doi.org/10.13140/RG.2.1.2194.5763
1999: https://doi.org/10.2166/wst.1999.0244
In the introduction section, I suggest you discuss and in particular focus on the economic evaluation aspects of these indexes.
- We thank the reviewer for sharing the papers on sanitation sustainability indexes. However, most of these references do not examine the economics or financial aspects. To address the reviewer’s suggestion, we have modified the specific phrase that the reviewer highlighted as follows:
- Lines 32-34: “Understanding household demand for improved sanitation infrastructure in these settings is one critical element of urban sanitation development efforts.”
The second concept is about the core of this paper, WTP. This study is focusing on the implementation of flush latrines. Again the authors have wisely discussed the role of "providing more explanation of sanitation benefits" on the "recommendations to increase voucher redemption." This is great. However, implementing a sanitation system that can give benefits to the community as well as the background of the community to utilize the human sanitary wastes as fertilizer may also be useful in increasing the WTP. In another word, the user may be willing to pay more if s/he knows that the system can provide him/her direct economic benefits.
In this regard, I wish to invite the authors to check the papers listed below regarding the direct benefit that a sanitation system can give to the community. This concept might be worthy to be discussed in the discussion section.
https://doi.org/10.1177/0734242X10390073
- Thank you for the suggestion of presenting households with the potential benefits of using human sanitary wastes as fertilizer. However, the paper that the reviewer shared does not make a strong case for convincing people to invest in sanitation for this reason, stating in the abstract “It was found that there is a general negative attitude to fresh excreta and the handling of it. However, the residents accept that excreta can be used as fertilizer, but they are not willing to use it on their own crops or consume crops fertilized with excreta.” We do not believe that there is good evidence that demand would increase if households knew that waste was going to be converted into fertilizers, especially because urban households are not the ones using the fertilizer.
- However, Nakuru has been a place of sanitation innovation, and we have added to the discussion:
- Lines 502-504: “In partnership with other stakeholders, NAWASSCO continues to explore innovative approaches to increase sanitation coverage in Nakuru, such as turning human waste into a resource [42].”